# Extraordinary pseudocapacitive energy storage triggered by phase transformation in hierarchical vanadium oxides

Bo-Tian Liu[1], Xiang-Mei Shi[1], Xing-You Lang [1], Lin Gu[2], Zi Wen[1], Ming Zhao[1] & Qing Jiang[1]

Pseudocapacitance holds great promise for improving energy densities of electrochemical supercapacitors, but state-of-the-art pseudocapacitive materials show capacitances far below their theoretical values and deliver much lower levels of electrical power than carbon-based materials due to poor cation accessibility and/or long-range electron transferability. Here we show that in situ corundum-to-rutile phase transformation in electron-correlated vanadium sesquioxide can yield nonstoichiometric rutile vanadium dioxide layers that are composed of highly sodium ion accessible oxygen-deficiency quasi-hexagonal tunnels sandwiched between conductive rutile slabs. This unique structure serves to boost redox and intercalation kinetics for extraordinary pseudocapacitive energy storage in hierarchical isomeric vanadium oxides, leading to a high specific capacitance of ~1856 F g$^{-1}$ (almost sixfold that of the pristine vanadium sesquioxide and dioxide) and a bipolar charge/discharge capability at ultrafast rates in aqueous electrolyte. Symmetric wide voltage window pseudocapacitors of vanadium oxides deliver a power density of ~280 W cm$^{-3}$ together with an exceptionally high volumetric energy density of ~110 mWh cm$^{-3}$ as well as long-term cycling stability.

[1] Key Laboratory of Automobile Materials (Jilin University), Ministry of Education, and School of Materials Science and Engineering, Jilin University, Changchun 130022, China. [2] Beijing National Laboratory for Condensed Matter Physics, The Institute of Physics, Chinese Academy of Sciences, Beijing 100190, China. These authors contributed equally: Bo-Tian Liu, Xiang-Mei Shi. Correspondence and requests for materials should be addressed to X.-Y.L. (email: xylang@jlu.edu.cn) or to Q.J. (email: jiangq@jlu.edu.cn)

Transition-metal oxides (TMOs) are attractive pseudocapacitive materials in electrochemical supercapacitors because they have theoretical-specific capacitance more than one or two orders of magnitude higher than that of carbon-based materials ($\sim$5–15 $\mu$F cm$^{-2}$)[1–3]. Unlike carbon materials in which only electric double-layer capacitance is available[1–6], pseudocapacitive TMOs store/release charges by cation adsorption/desorption (redox pseudocapacitance)[7–9] or/and intercalation/de-intercalation (intercalation pseudocapacitance)[10–13] coupled with reversible redox reactions of metal ions at or near the electrode/electrolyte interface. Both faradaic mechanisms can work separately or together, depending on the crystallographic structures of the electroactive materials whether they localize the redox reactions on the surface or accommodate cations in the interlayer gaps or tunnels[2, 14, 15]. However, no matter which type of pseudocapacitive mechanisms is mainly involved in the charge storage/delivery, the electroactive TMOs are required to play dual roles during the charge/discharge processes, i.e., accommodating cations at the surface or in the interlayer gaps, and transferring the generated electrons from the redox sites to the conductive materials[16]. The tradeoff between cation accessibility/diffusion and electronic conductivity in state-of-the-art TMOs makes them difficult to really realize the energy storage with battery-like capacity and carbon-based supercapacitors-like rate performance. For instance, the redox pseudocapacitive TMOs (such as RuO$_2$[17, 18] and MnO$_2$[19–21]) often have too small interlayer gaps to accommodate cation insertion/extraction, which essentially localizes the redox reaction of metal ions at the surface of electroactive materials on the time scale of interest[21, 22]; while the ones with facile cation accessibility and diffusion (such as layered Nb$_2$O$_5$[11], MnO$_2$[22] and MoO$_3$[23]) generally suffer from poor electronic conductivity[11, 15, 16, 21–24], which significantly impedes the long-range transfer of electrons from the surface redox sites to current collectors. Consequently, they usually achieve practical capacitances far below their theoretical expectations and deliver much lower levels of electrical power than carbon materials, unsatisfying the fast-growing demands of high-power and high-energy densities in portable electronic devices and hybrid vehicles with limited area and volume.

Here we report a class of bipolar TMOs, of which the crystallographic structures are designed and regulated to facilitate simultaneously cation accessibility/diffusion and electron transfer, for realizing high levels of energy storage at fast charge/discharge rates in symmetric aqueous pseudocapacitors with a wide voltage window. Specifically, vacancy-ordered rutile VO$_{2-x}$ phases (r-VO$_{2-x}$), which are composed of highly cation accessible hexagonal oxygen-deficiency tunnels and conductive r-VO$_2$ ($x = 0$) slabs, are in situ produced and seamlessly integrated with the precursor corundum V$_2$O$_3$ (c-V$_2$O$_3$) core in a hierarchically nanoporous architecture (NP c-V$_2$O$_3$/r-VO$_{2-x}$) by a thermal-oxidation-triggered corundum-to-rutile (CTR)-phase transformation. As a result of offering both intercalation and redox pseudocapacitance with similarly facile kinetics and shortening electron transfer distance from the redox sites to conductive intermediate via metallic V–V chains, the NP c-V$_2$O$_3$/r-VO$_{2-x}$ hybrid electrodes exhibit about six times the specific and volumetric pseudocapacitance of NP r-VO$_2$ ($\sim$1856 F g$^{-1}$ and $\sim$1933 F cm$^{-3}$) with exceptionally high-rate performance. This enlists their aqueous pseudocapacitors to have a volumetric energy of $\sim$330 mWh cm$^{-3}$ ($\sim$13 mWh cm$^{-3}$ based on the whole volume of device, comparable to that of 4 V/500 $\mu$Ah thin-film lithium batteries) at high levels of power delivery similar to carbon-based supercapacitors.

## Results

**DFT simulation.** Both vanadium sesquioxide and dioxide are archetypal electron-correlated materials with metal-to-insulator transitions (MITs)[25, 26], through which their high-temperature phases, i.e., c-V$_2$O$_3$ and r-VO$_2$ (Supplementary Figure 1a, b), are conductive in virtue of the itinerant 3d$^2$ (V$^{3+}$) and 3d$^1$ (V$^{4+}$) electrons along short V–V chains, respectively. As illustrated by typical temperature-resistivity curves for V$_2$O$_3$ and VO$_2$ films deposited on Al$_2$O$_3$ substrates, their electronic conductivity reaches $\sim$10$^3$–10$^4$ s cm$^{-1}$ at room temperature by virtue of metallization via MITs (Supplementary Figure 1c). Distinguished from the rhombohedral c-V$_2$O$_3$ with the tetrahedral interstices that are too small to assist Na ion diffusion[27, 28], the r-VO$_2$ has a tunnel structure, in which both cations and electrons prefer to transport along the z-axis tunnels and the shortly V–V bonded walls, respectively[27, 29, 30]. These distinct properties imply their favorable roles in electrochemical energy storage: the former serving as the conductor to facilitate electron transport along a three-dimensional (3D) vanadium-atom framework and the latter accommodating cation storage in the tunnels as the electroactive intermediate. Nevertheless, the pristine r-VO$_2$ with [1×1] tunnels exhibits unexpectedly low capacitance because of unnegligible oscillation of electrostatic potential near the surface[31], which dramatically increases the initial energy barrier and thus confines the redox reaction at the electrode/electrolyte interface[31, 32]. Using density functional theory (DFT) calculations, we demonstrate that the CTR-phase transformation in oxygen nonstoichiometry produces the r-VO$_{2-x}$ phase to grow epitaxially on the precursor c-V$_2$O$_3$ substrate on account of the compressive strain at their interface (Fig. 1a and Supplementary Figure 2). This metastable rutile phase consists of quasi-hexagonal tunnels with a cross-section area of $\sim$24 Å$^2$ alternatingly sandwiched between rutile slabs with thicknesses of unit cells (Fig. 1b and Supplementary Figure 3). Therein, Na ions tend to be levitated at the tunnel center with the intercalation energy ($E_{int}$) of about $-0.8$ eV and transport along the tunnels with a low energy barrier ($E_b$) of $\sim$0.02 eV because of their analogous interactions with the surrounding oxygen atoms (Supplementary Figure 4a), in sharp contrast to those in the pristine r-VO$_2$ with $E_{int} = \sim$0.9 eV and $E_b = \sim$0.1 eV (Fig. 1c, d). Meanwhile, not only the rutile slabs but also the walls of quasi-hexagonal tunnels remain high electronic conductivity, which enables electron transfer in the atomic distance from the redox sites to the conductive V–V chains[27, 29] (Fig. 1e). As shown in the typical V–V chains along z-axis (Fig. 1f, g), the oxygen nonstoichiometry results in evident orbital overlapping between the nearest neighbor V atoms, where the charge densities associated with states close to Fermi level are calculated according to the density of states ranging from $E_F$-0.8 to $E_F$+0.8 eV (Supplementary Figure 4b). The increase of rutile slab thickness does not give rise to remarkable changes of electron and ion transport kinetics except for the influence on the capacitance of r-VO$_{2-x}$ (Fig. 1c, d), in which the quasi-hexagonal tunnels accommodate Na$^+$ ions to offer intercalation pseudocapacitance ($C_{v,int}$) and the r-VO$_2$ slabs localize the redox reaction on the surface for the redox pseudocapacitance ($C_{s,VO2}$). As shown in the phase diagram for specific capacitance versus x value (Supplementary Figure 5), the r-VO$_{2-x}$ is predicted to exhibit a linearly increasing specific capacitance with the increasing x value and reach the theoretical value (2292 F g$^{-1}$) at $x = 1/4$ according to the equation, $C_s = [4xC_{v,int} + (1-4x)C_{s,VO2}]$. As the x value further increases, there forms a mixture of r-VO$_{2-x}$ and c-V$_2$O$_3$ with the decreasing specific capacitance, which is determined in terms of

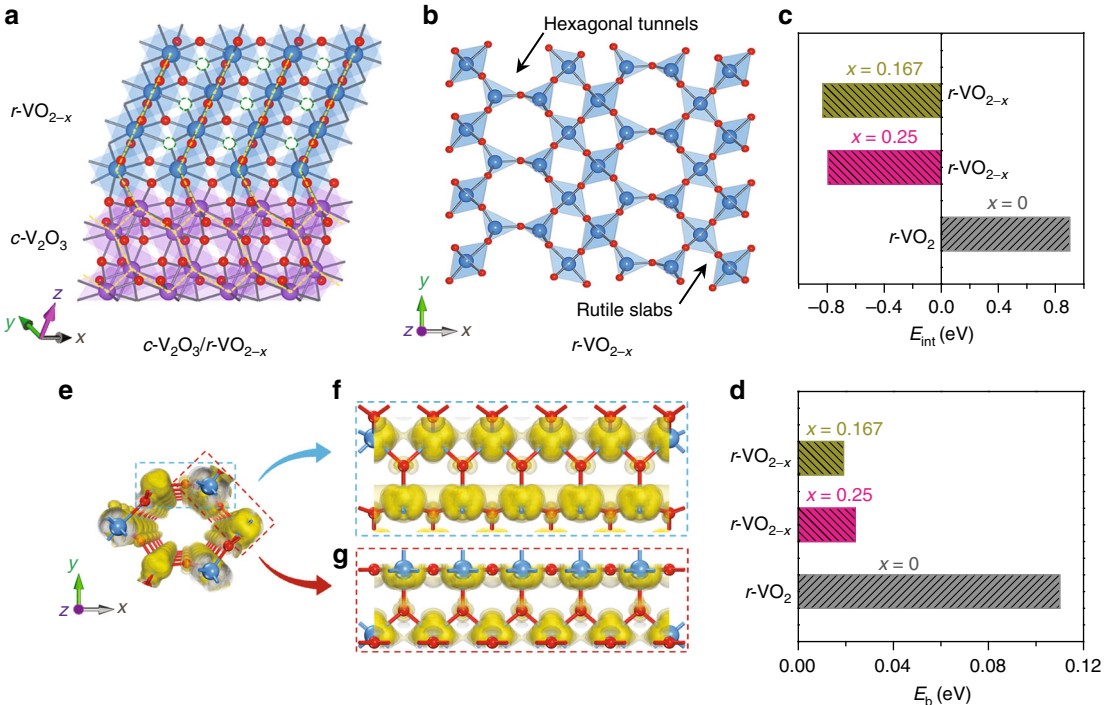

**Fig. 1** Structures and DFT computations of vanadium oxides with high electronic conductivity and ionic accessibility. **a** Atomic schematic illustration for the isomeric vanadium oxides produced by in situ corundum-to-rutile-phase transformation, which consist of corundum-type $V_2O_3$ and rutile $r$-$VO_{2-x}$ core/shell structure. Yellow dashed lines denote the electron transport pathways along short V–V bonds. Purple and blue balls represent vanadium atoms in corundum-type $V_2O_3$ and rutile-type $VO_{2-x}$, red and white ones denote oxygen atoms and ordered oxygen vacancies. **b** Representative atomic structure of the $r$-$VO_{2-x}$ layer with ordered 1D quasi-hexagonal tunnels along $z$-axis. **c**, **d** Comparison of intercalation energy ($E_{int}$) (**c**) and energy barrier ($E_b$) (**d**) for Na$^+$ in the tunnels of the $r$-$VO_2$ and $r$-$VO_{2-x}$ with $x = 0.25$ and 0.167. **e–g** Electron density distributions of the projected orbitals for the quasi-hexagonal tunnel (**e**) and the V-O walls of quasi-hexagonal tunnel (**f**) and rutile slab (**g**)

$C_s = \{4(0.5-x)C_{v,int} + [1-4(0.5-x)]C_{s,V2O3}\}$. Here $C_{s,V2O3}$ is the redox pseudocapacitance of $c$-$V_2O_3$.

**Synthesis and characterizations**. To extend the DFT predictions to designing practical electrodes, the NP $c$-$V_2O_3$/$r$-$VO_{2-x}$ hybrids are fabricated by a facile thermal oxidation of $c$-$V_2O_3$ precursor skeletons under the short supply of oxygen (Fig. 2a). Therein, the in situ CTR-phase transformation is triggered to produce the $r$-$VO_{2-x}$ layers (Fig. 2b), which evolve with the thermal-oxidation time. Figure 2c and Supplementary Figure 6a show typical scanning electron microscope (SEM) images of NP $c$-$V_2O_3$/$r$-$VO_{2-x}$ film electrodes with thickness of ~1.2 μm (thermal oxidation, 10 min), displaying the same 3D bicontinuous nanoporous structure as that of the NP $c$-$V_2O_3$ skeleton which consists of periodic walls and multimodal open nanopores. In addition to the ordered macropores with sizes of 360 nm and ~100 nm based on opal template, there are abundant mesopores and micropores to be directly observed in the transmission electron microscope (TEM) images of NP $c$-$V_2O_3$/$r$-$VO_{2-x}$ walls (Fig. 2d and Supplementary Figure 6b, c). The multimodal feature is confirmed by a type IV nitrogen adsorption/desorption isotherm (Supplementary Figure 6d), which signifies a mesopore size distribution with distinct maxima centered at ~4 nm, ~10 nm and ~70 nm (Supplementary Figure 6e). The ultrasmall pores in the NP $c$-$V_2O_3$/$r$-$VO_{2-x}$ film electrodes are responsible for the Brunauer–Emmett–Teller (BET) surface areas of as high as ~99.8 m$^2$ g$^{-1}$, and the large pores provide electrolyte channels to enhance ion transport properties. During the CTR-phase transformation, the shearing movement of V atoms in the $c$-$V_2O_3$(012) basal plane along [12$\bar{1}$] direction gives rise to the formation of $r$-

$VO_2$(011) planes but with a mismatch of 8.08% along the [100] direction[33, 34] (Supplementary Figures 2a and 7). Such large compressive strain makes it thermodynamically favorable to form ordered oxygen vacancies, i.e., the quasi-hexagonal tunnels in the $r$-$VO_{2-x}$ phases, for lowering elastic energy (Supplementary Figure 7) when they are epitaxially growing on the $c$-$V_2O_3$ skeleton along the [001] direction (Supplementary Figure 2b, c). This is verified by a compelling evidence demonstrated by high-resolution transmission electron microscope (HR-TEM) and aberration-corrected high-angle annular dark-field (HAADF) scanning TEM (HAADF-STEM). As shown in representative HR-TEM images of $c$-$V_2O_3$/$r$-$VO_{2-x}$ (Fig. 2e), the metastable $r$-$VO_{2-x}$($\bar{1}$01) layer is seamlessly integrated with the $c$-$V_2O_3$(110) core skeletons via short V–V bonds (Fig. 2f and Supplementary Figure 2d). Viewed along the [001] axis, the structure composed of alternating quasi-hexagonal tunnels and rutile slabs with [1×1] tunnels is directly observed from atomic-resolution HAADF-STEM image (Fig. 2g), wherein the atoms sitting inside the hexagonal tunnels are attributed to the V atoms in the $V_2O_3$ substrate. Further image simulations confirm that bright spots correspond to atomic columns, among which the dark hexagons and squares are due to the quasi-hexagonal tunnels and [1×1] tunnels in the rutile slabs, respectively (Fig. 2h). A typical line profile demonstrates a large and aperiodic fluctuation around ~1 nm to validate the quasi-hexagonal tunnel that is produced in the CTR transformation via shearing movement (Fig. 2i).

Such CTR-phase transformation process is also attested by X-ray diffraction (XRD) patterns (Fig. 3a) and Raman spectra (Supplementary Figure 8a) for the NP $c$-$V_2O_3$/$r$-$VO_{2-x}$ films, of which the characteristic peaks match well with those of the pristine NP $c$-$V_2O_3$ and $r$-$VO_2$ films. As shown in the XRD

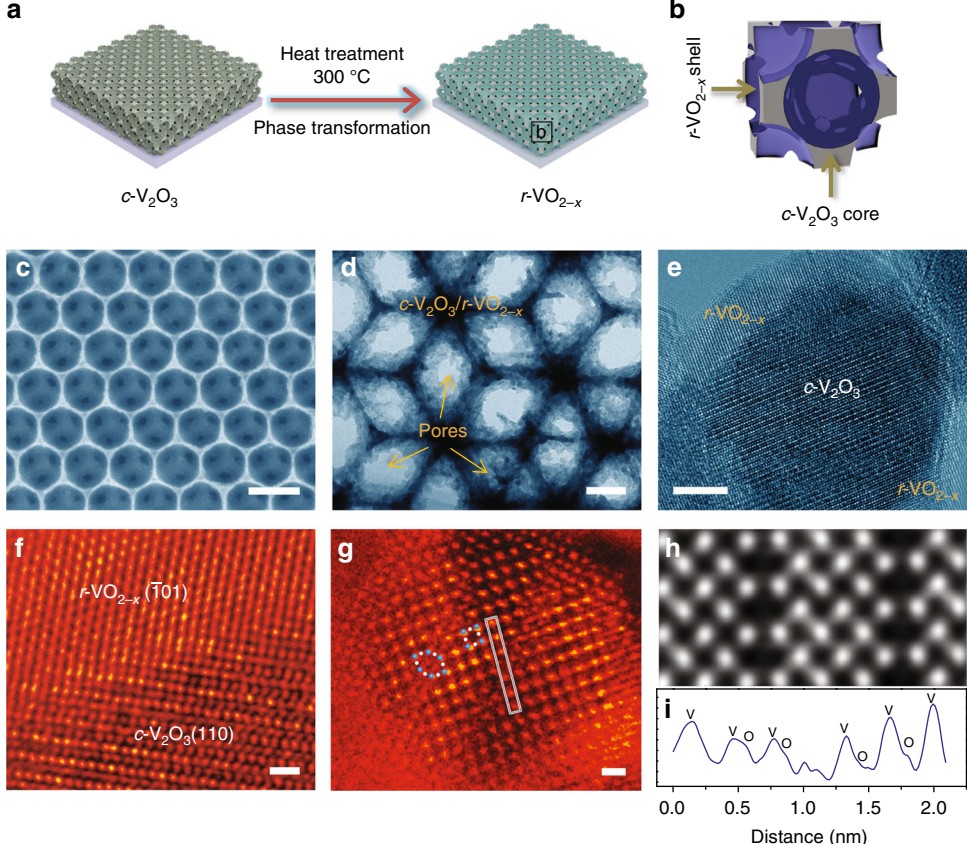

**Fig. 2** Preparation and structural characterization of isomeric vanadium oxides. **a** Scheme for in situ corundum-to-rutile-phase transformation triggered by thermal oxidation to fabricate hierarchical nanoporous isomeric $c$-$V_2O_3$/$r$-$VO_{2-x}$ from precursor 3D $V_2O_3$ skeleton. **b** Core/shell structure of isomeric of $c$-$V_2O_3$/$r$-$VO_{2-x}$. **c** SEM image of nanoporous $c$-$V_2O_3$/$r$-$VO_{2-x}$ films with an ordered macroporous structure. Scale bar, 500 nm. **d–f** Bright-field TEM and HR-TEM images of $c$-$V_2O_3$/$r$-$VO_{2-x}$ core/shell structure (**d**, **e**) and epitaxially interfacial structure (**f**). Scale bar, 100 nm (**d**), 5 nm (**e**) and 1 nm (**f**). **g** HAADF-STEM image of the $r$-$VO_{2-x}$ layer viewed from the [001] axis, demonstrating an alternating structure of quasi-hexagonal tunnels and rutile slabs. Scale bar, 0.5 nm. **h** Simulated HAADF-STEM image of quasi-hexagonal tunnel structure in (**g**). **i** Line profile of column of atoms in the white boxed area

pattern of the NP $c$-$V_2O_3$/$r$-$VO_{2-x}$ film (thermal oxidation, 10 min), the diffraction peaks correspond to the (110), (011), (111), (120), (121), (220) and (130) planes of the rutile phase in space group $P4_2/mnm$ (JCPDS 44–0253), apart from ones attributed to the precursor $c$-$V_2O_3$ (JCPDS 34–0187). These characteristic peaks of the $r$-$VO_{2-x}$ gradually intensify with the thermal-oxidation time from 10 to 60 min, indicating the increase of rutile-phase volume fraction. While extending the thermal-oxidation time to 90 min, the NP $c$-$V_2O_3$ is completely transformed into the NP $r$-$VO_2$ (Fig. 3a and Supplementary Figure 8a). The evolution of chemical state of V atoms in the $r$-$VO_{2-x}$ layer during the CTR-phase transformation is identified by X-ray photoelectron spectroscopy (XPS). Compared with the XPS spectrum of the NP $c$-$V_2O_3$ film mainly containing $V^{3+}$ (Fig. 3b), there are only V $2p_{3/2}$ (516.6 eV) and V $2p_{1/2}$ (524.1 eV) peaks corresponding to $V^{4+}$ in the NP $r$-$VO_2$ (Fig. 3d)[35]. While in the XPS spectra of the NP $c$-$V_2O_3$/$r$-$VO_{2-x}$ films, the V $2p_{3/2}$ and V $2p_{1/2}$ signals are deconvoluted into two more peaks at the binding energies of 515.5 and 522.8 eV due to the presence of $V^{3+}$ in the quasi-hexagonal tunnels[35], in addition to those of $V^{4+}$ in the rutile slabs (Fig. 3c and Supplementary Figure 9). According to the intensity ratio of $V^{3+}$/$V^{4+}$ at the V $2p_{3/2}$ peaks, the $x$ value in the $r$-$VO_{2-x}$ is determined to be in the range from 0.22 to 0.1 with the thermal-oxidation time (Supplementary Figure 10). Such high density of quasi-hexagonal tunnels in the $r$-$VO_{2-x}$ layer not only ameliorates the cation accessibility but also stabilizes the conductive rutile phases at room temperature by shifting the MIT

temperature down to ~180 K (Supplementary Figure 1c)[26]. In particular, almost all the V–V bonds at their interfaces is shorter than the critical separation value of the 3$d$ electron coupling interaction (0.293 nm) (Supplementary Figure 11)[27], which serve as the electron transport pathways between $V_2O_3$ and $r$-$VO_{2-x}$ without any additional contact resistance. As illustrated by the current–voltage (I–V) measurement on the NP $c$-$V_2O_3$/$r$-$VO_{2-x}$ film ($x = 0.22$), the I–V curve in the range of −0.8 to 0.8 V is linear (Fig. 3e), with a resistance of ~17 Ω, only slightly higher than that of the pristine NP $c$-$V_2O_3$ (~14 Ω) (inset of Fig. 3e).

**Electrochemical characterizations.** The electrochemical properties of nanoporous vanadium oxide electrodes are measured in a three-electrode configuration using a Pt foil as the counter electrode and a Ag/AgCl electrode as reference electrode. Figure 4a shows typical cyclic voltammograms (CVs) for the NP $c$-$V_2O_3$/$r$-$VO_{2-x}$ ($x = 0.22$), $c$-$V_2O_3$ and $r$-$VO_2$ electrodes in 1 M $Na_2SO_4$ aqueous electrolyte at a scan rate of 50 mV s$^{-1}$, exhibiting a quasi-rectangular shape in a potential window of 0 to 0.8 V. In virtue of the 3D hierarchical and bicontinuous nanoporous architecture which simultaneously provides high electron and ion transport pathways, their CV curves at various scan rates from 5 to 1000 mV s$^{-1}$ retain a quasi-rectangular shape, indicating their extraordinary high-rate performance (Supplementary Figure 12a,b,c). Relative to the NP $c$-$V_2O_3$ and $r$-$VO_2$ electrodes, the NP $c$-$V_2O_3$/ $r$-$VO_{2-x}$ hybrid electrode has remarkably enhanced current

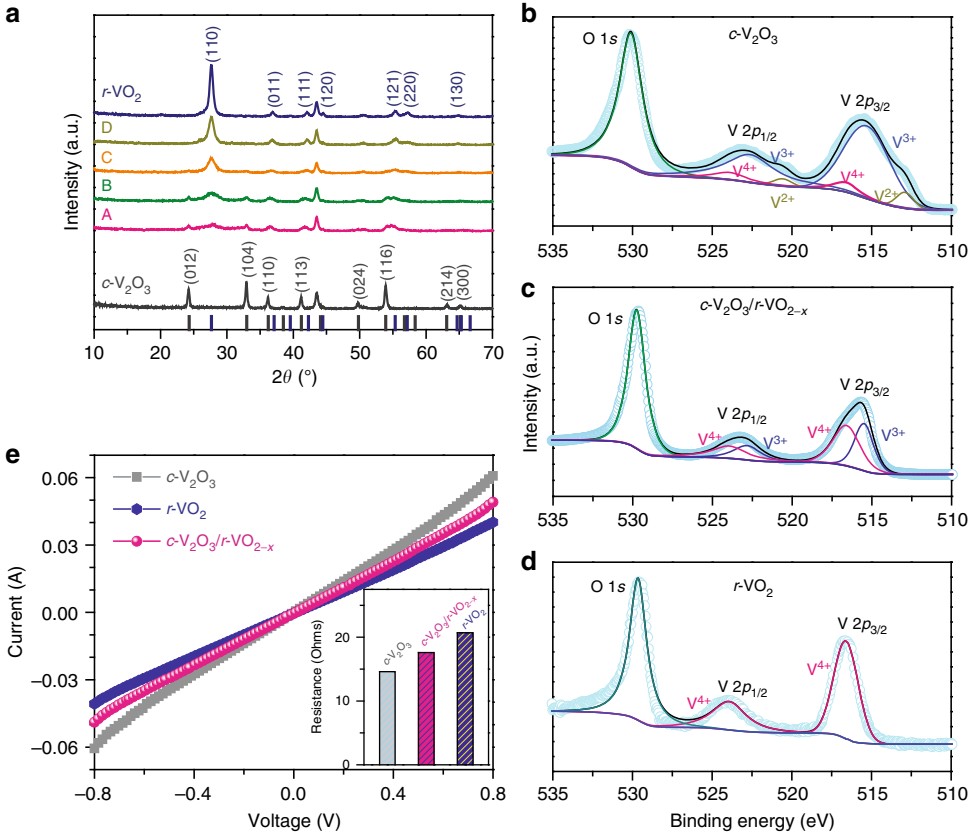

**Fig. 3** Chemical analysis and electrical properties. **a** XRD patterns for NP $c$-$V_2O_3$, $r$-$VO_2$ and $c$-$V_2O_3$/$r$-$VO_{2-x}$ films, demonstrating the evolution of phase transformation via thermal oxidation of the corundum $c$-$V_2O_3$ at 300 °C for 10 (A), 20 (B), 30 (C) and 60 min (D), respectively. The line patterns show reference cards 34–0187 and 44–0253 for the pristine $c$-$V_2O_3$ and $r$-$VO_2$ according to JCPDS. **b**–**d** XPS spectra of O 1s, and V 2p for NP $c$-$V_2O_3$ (**a**), $c$-$V_2O_3$/$r$-$VO_{2-x}$ (**b**) and $r$-$VO_2$ films (**c**), which are synthesized by thermal oxidation of the as-prepared $c$-$V_2O_3$ films for 0, 10, 90 min at 300 °C, respectively. **e** I–V curves and the corresponding resistances (inset) of NP $c$-$V_2O_3$, $c$-$V_2O_3$/$r$-$VO_{2-x}$ ($x = 0.22$) and $r$-$VO_2$ films

density because of the distinguished $r$-$VO_{2-x}$ layer, in which the quasi-hexagonal tunnels facilitate the $Na^+$ intercalation/de-intercalation in addition to the high conductivity of V–V chains for the fast, reversible surface redox reactions for more energy storage/delivery. This is substantiated by the galvanostatic charge/discharge measurements with a current density of 10.4 A cm$^{-3}$, at which it takes much longer time to charge/discharge the NP $c$-$V_2O_3$/$r$-$VO_{2-x}$ than the NP $c$-$V_2O_3$ and $r$-$VO_2$ (Supplementary Figure 13). Figure 4b compares the gravimetric and volumetric capacitances of NP $c$-$V_2O_3$/$r$-$VO_{2-x}$ ($x = 0.22$) with the values of the NP $c$-$V_2O_3$ and $r$-$VO_2$ at various scan rates (Supplementary Note 1). The NP $c$-$V_2O_3$/$r$-$VO_{2-x}$ electrode achieves a gravimetric capacitance of as high as ~1856 F g$^{-1}$ (corresponding to the volumetric capacitance of ~1933 F cm$^{-3}$) at a scan rate of 5 mV s$^{-1}$. When the scan rate is increased to 1000 mV s$^{-1}$, it still retains the capacitance of 760 F g$^{-1}$ or 792 F cm$^{-3}$, about 20 times the values of the NP $r$-$VO_2$ (36 F g$^{-1}$ or 37 F cm$^{-3}$). This exceptional rate performance enlists the NP $c$-$V_2O_3$/$r$-$VO_{2-x}$ electrode to outperform not only volumetrically but gravimetrically some of the best pseudocapacitive electrodes in a full rate range reported previously: such as nanotubular arrayed RuO$_2$[9], hydrogenated-TiO$_2$/MnO$_2$ (H-TiO$_2$/MnO$_2$) hybrid[19], bare MnO$_2$[20], Ti$_3$C$_2$T$_x$ MXene clay[36], N-doped mesoporous few-layer carbon (MFLC-N)[37] and nanostructured hexagonal WO$_3$[38]. Even for the NP $c$-$V_2O_3$/$r$-$VO_{2-x}$ film electrode with thickness increasing to 7.8 μm, almost same pseudocapacitive behavior remains (Supplementary Figure 14).

To illustrate the unique charge/discharge kinetics, the anodic/cathodic current ($i$) is assumed to obey a power-law relationship

of scan rate ($v$)[10, 11, 15, 23], i.e., $i = av^b$, with $a$ and $b$ being adjustable values. The $b$-value of 0.5 or 1 indicates that the current is a diffusion- or surface-controlled process, respectively. In a ln($i$)–ln($v$) plot, the NP $c$-$V_2O_3$/$r$-$VO_{2-x}$ ($x = 0.22$) electrode possesses a $b$-value of 1 in the scan rates ranging from 5 to 100 mV s$^{-1}$, implying the surface-controlled kinetics in the discharging time >8 s. While for $v > 100$ mV s$^{-2}$, the $b$-value decreases to 0.76, which is due to the kinetic constraint of $Na^+$ diffusion in addition to ohmic contribution revealed by electrochemical impedance spectroscopy analysis in a frequency range from 100 kHz to 10 mHz (Supplementary Figure 15 and Supplementary Note 2)[39, 40]. This is in sharp contrast with the charge storage in pristine NP $c$-$V_2O_3$ and $r$-$VO_2$, which is dominated by a diffusion-controlled process in a full range of scan rates (Supplementary Figure 12d)[10, 11, 15]. Owing to the remarkably enhanced kinetics of $Na^+$ accessibility and electron transport, the interfacial charge transfer resistance ($R_{CT}$) and the Warburg resistance ($Z_w$) of the NP $c$-$V_2O_3$/$r$-$VO_{2-x}$ electrode are much lower than the values of the NP $c$-$V_2O_3$ and $r$-$VO_2$ (Supplementary Figure 15c, d). Figure 4c shows the plot of normalized capacitance versus $v^{-1/2}$ for the NP $c$-$V_2O_3$/$r$-$VO_{2-x}$ ($x = 0.22$) electrode from 5 to 1000 mV s$^{-1}$. Two distinct regions in $v < 100$ mV s$^{-1}$ and $v > 100$ mV s$^{-1}$ are exhibited, further validating the fact that the total specific capacitance ($C_s$) of NP $c$-$V_2O_3$/$r$-$VO_{2-x}$ arises from the diffusion-controlled intercalation pseudocapacitance (rate-dependent component, $\kappa_2 v^{-1/2}$) coupled with the surface-controlled redox pseudocapacitance (rate-independent component, $\kappa_1$)[10, 11, 15], i.e., $C_s = \kappa_1 + \kappa_2 v^{-1/2}$. At scan rates below 100 mV s$^{-1}$, the extrapolated intercept to $v^{-1/2} = 0$ yields

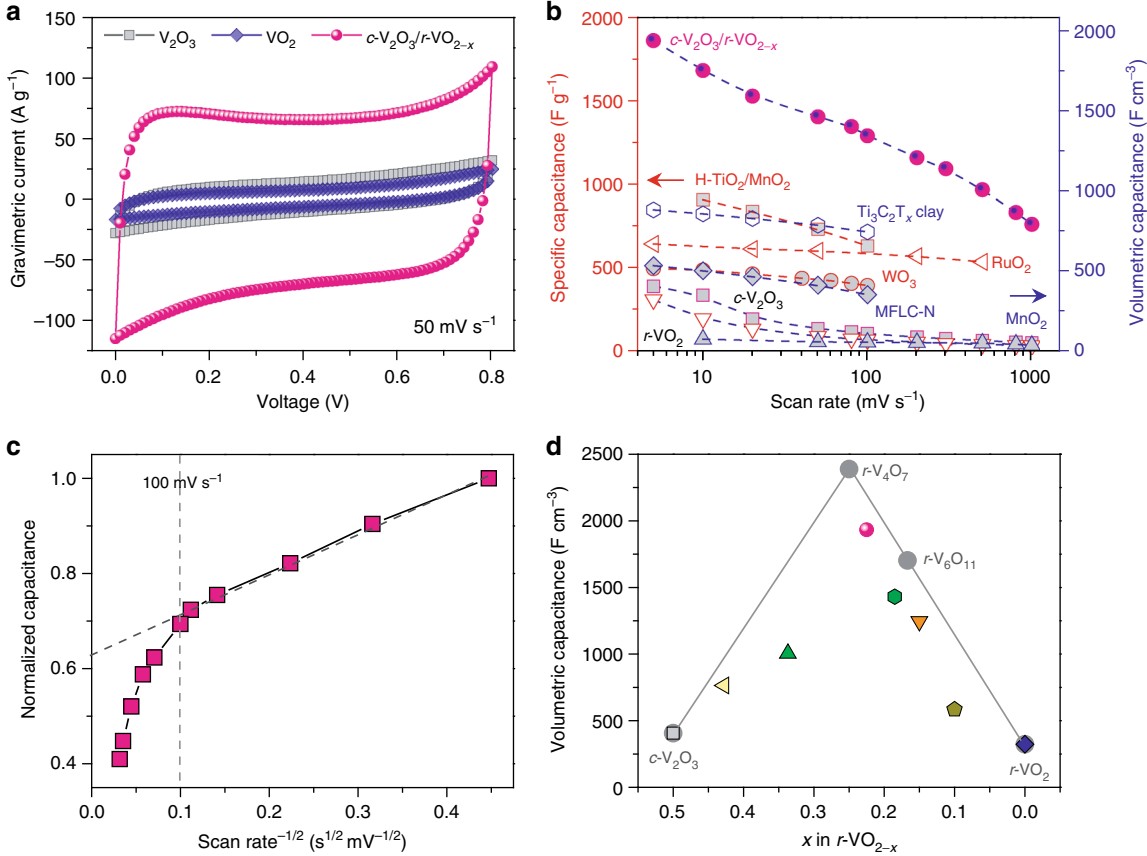

**Fig. 4** Electrochemical characterization of electrodes in aqueous electrolyte. **a** Cyclic voltammetry (CV) curves of NP $c$-$V_2O_3$, $r$-$VO_2$ and $c$-$V_2O_3$/$r$-$VO_{2-x}$ ($x = 0.22$) electrodes in three-electrode configuration at a scan rate of 50 mV s$^{-1}$ in 1 M $Na_2SO_4$. **b** Gravimetric and volumetric capacitances for NP $c$-$V_2O_3$, $r$-$VO_2$ and $c$-$V_2O_3$/$r$-$VO_{2-x}$ electrodes at various scan rates, comparing with the volumetric values previously reported for $Ti_3C_2T_x$ MXene clay[36], N-doped mesoporous few-layer carbon (MFLC-N)[37] and bare $MnO_2$ electrodes[20], as well as the gravimetric values of mesoporous $RuO_2$[9], hydrogenated-$TiO_2$/$MnO_2$ (H-$TiO_2$/$MnO_2$)[19] and nanostructured hexagonal $WO_3$[38]. **c** Plot of normalized capacitance versus scan rate$^{-1/2}$ for the separation of diffusion-controlled intercalation pseudocapacitance from surface-controlled redox pseudocapacitance in the scan rates from 5 to 1000 mV s$^{-1}$. **d** Evolution of volumetric capacitances of NP $c$-$V_2O_3$/$r$-$VO_{2-x}$ electrodes at a scan rate of 5 mV s$^{-1}$ as a function of $x$. The gray solid circles and lines are the theoretically volumetric capacitances for NP $c$-$V_2O_3$/$r$-$VO_{2-x}$ based on the atomic structures with $x = 0$, 0.167, 0.25 and 0.5, which correspond to $r$-$VO_2$, metastable $r$-$VO_{2-x}$ and $c$-$V_2O_3$, respectively

the redox pseudocapacitance that just accounts for about two fifths of total capacitance. This indicates that the charge storage is mainly contributed by the $Na^+$ insertion/extraction, which behaves in a capacitive energy storage for the charge/discharge time of 8 s or longer. Whereas the intercalation pseudocapacitance is conventionally limited by solid-state diffusion in conventional TMOs[10, 11, 15], it improves significantly the energy-storage density without compromise of rate performance in the NP $c$-$V_2O_3$/$r$-$VO_{2-x}$ electrodes because of the unique microstructure of $r$-$VO_{2-x}$, which offers both redox and intercalation pseudocapacitance with similarly facile kinetics. The dominance of $Na^+$ intercalation/de-intercalation is further demonstrated by the fact that the NP $c$-$V_2O_3$/$r$-$VO_{2-x}$ electrodes maintain similar pseudocapacitive behaviors and rate performance while their volumetric capacitances increase with the $x$ value, but are independent of the production amount of rutile phase in the CTR-phase transformation (Fig. 4d and Supplementary Figure 16). This is also verified by the general observation for the NP $c$-$V_2O_3$ decorated with few $r$-$VO_{2-x}$ by shortening the thermal-oxidation time (Supplementary Figure 17), which exhibits slightly enhanced volumetric capacitance despite the negative deviation in the range of $x$ values far away from one fourth due to the contribution of the disordered distribution of quasi-hexagonal tunnels, i.e., the effect of configurational entropy

(Fig. 4d). Whereas vanadium oxides generally suffer from cycling instability, our NP $c$-$V_2O_3$/$r$-$VO_{2-x}$ electrode shows exceptional durability in long-term galvanostatic charge/discharge cycles (Supplementary Figure 18), which are performed in a potential window of 0–0.8 V at a current density of 80 A g$^{-1}$ in 1 M $Na_2SO_4$ aqueous electrolyte (inset of Supplementary Figure 18a). After 10,000 cycles, more than 90% capacitance still remained (Supplementary Figure 18a). XPS analysis demonstrates that the slight degradation of capacitance may result from the weeny change of surface valance state of $r$-$VO_{2-x}$ layer ($x = 0.23$) relative to that of the as-prepared one ($x = 0.22$) (Supplementary Figure 18b), but not the dissolution of vanadium oxides which usually takes place in previously reported vanadium oxide electrodes (Supplementary Figure 18c)[41, 42].

**Electrochemical performance of devices.** In view of the bipolar property of NP $c$-$V_2O_3$/$r$-$VO_{2-x}$ electrode (Supplementary Figure 19), symmetric pseudocapacitors are assembled with two identical NP $c$-$V_2O_3$/$r$-$VO_{2-x}$ films as both cathode and anode, and one piece of cotton paper as a separator for evaluating the practical energy-storage performance. By virtue of the special bipolar properties of vanadium oxides, their voltage windows can be extended to 1.4 V from 0.8 V in 1 M $Na_2SO_4$ aqueous

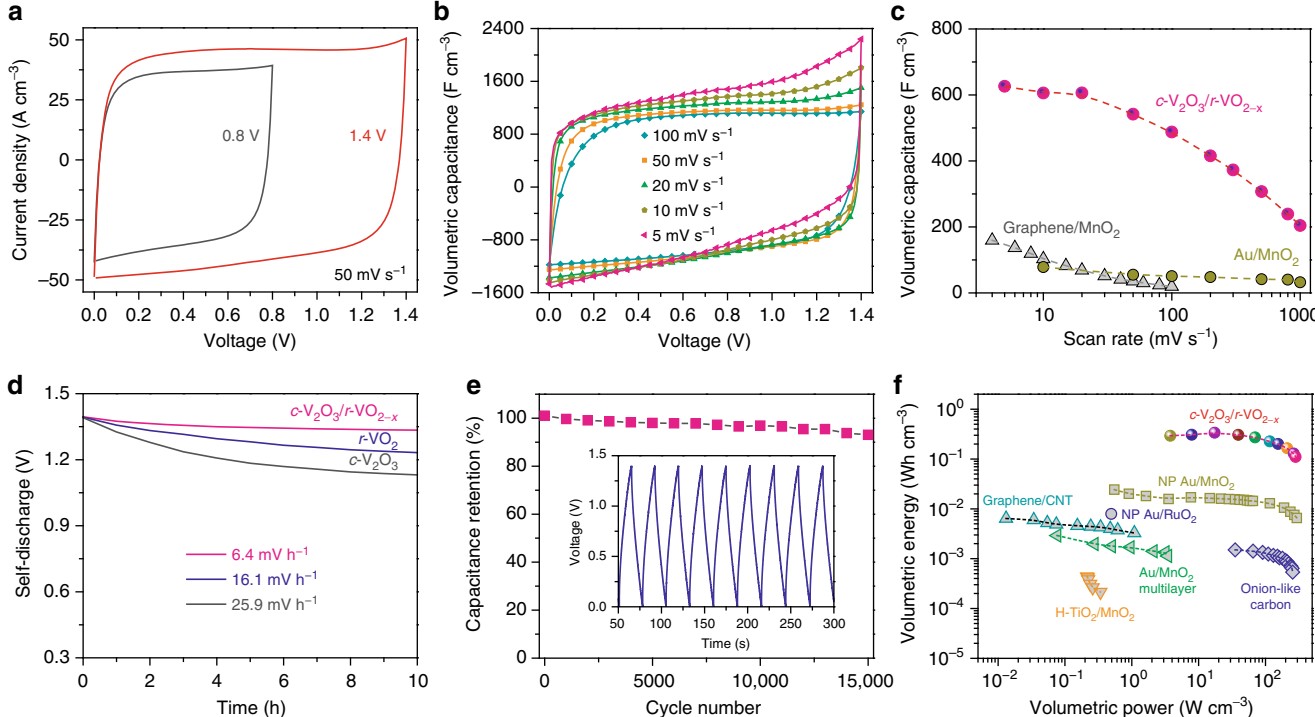

**Fig. 5** Performance of aqueous symmetric pseudocapacitor devices in a wide voltage window. **a** CV curves for symmetric pseudocapacitors, with NP $c$-$V_2O_3$/$r$-$VO_{2-x}$ ($x = 0.22$) as electrodes, in the voltage window extended from 0.8 to 1.4 V at a scan rate of 50 mV s$^{-1}$ in 1 M $Na_2SO_4$ aqueous electrolyte. **b** CV profiles in a voltage window of 1.4 V for pseudocapacitors at different scan rates. **c** Volumetric capacitances of device of NP $c$-$V_2O_3$/$r$-$VO_{2-x}$ ($x = 0.22$) electrode at various scan rates, comparing with the volumetric values of previously reported supercapacitors based on graphene/$MnO_2$ hybrid electrodes and Au/$MnO_2$ multilayers. **d** Self-discharge performances for symmetric pseudocapacitors based on NP $c$-$V_2O_3$/$r$-$VO_{2-x}$, $c$-$V_2O_3$ and $r$-$VO_2$ electrodes. **e** Cycling stability performance for NP $c$-$V_2O_3$/$r$-$VO_{2-x}$-based pseudocapacitor in the voltage window of 1.4 V. Inset: Galvanostatic cycling curves collected at 130 A cm$^{-3}$ in 1 M $Na_2SO_4$ aqueous electrolyte. **f** Ragone plot of volumetric power versus volumetric energy for pseudocapacitors based on total volumetric of NP $c$-$V_2O_3$/$r$-$VO_{2-x}$ in two electrodes, comparing with other electrochemical capacitors based on representative electrode materials such as onion-like carbon[5] and graphene/CNT hybrid[6], as well as H-$TiO_2$/$MnO_2$[19], Au/$MnO_2$ multilayers[20], NP Au/$MnO_2$ and NP Au/$RuO_2$[44]

electrolyte when securing against the occurrence of oxygen evolution reaction. As shown in Fig. 5a, the CV curve of symmetric NP $c$-$V_2O_3$/$r$-$VO_{2-x}$ ($x = 0.22$) pseudocapacitor at a scan rate of 50 mV s$^{-1}$ displays a perfect rectangular and symmetrical shape within the wide voltage of 1.4 V because of the identical pseudocapacitance and charge/discharge kinetic properties of two electrodes. Furthermore, their facile redox reaction and cation intercalation kinetics not only realizes high-density charge storage but also leads to exceptional high-rate performance of the symmetric pseudocapacitor over a wide range from 5 to 1000 mV s$^{-1}$. Therein, the Na$^+$ intercalation is demonstrated by Raman spectra of the charged/discharged NP $c$-$V_2O_3$/$r$-$VO_{2-x}$ films (Supplementary Figure 20). There appears new characteristic Raman peaks at the frequencies of 962 and 224 cm$^{-1}$, in addition to the shoulders 420 and 162 cm$^{-1}$, for the charged NP $c$-$V_2O_3$/$r$-$VO_{2-x}$ film due to the Na$^+$ intercalation[43]. Although the pseudocapacitor starts to exhibit resistive behavior at 200 mV s$^{-1}$, the symmetric CV retains quasi-rectangular shape even at the scan rate increasing to 1000 mV s$^{-1}$ (Fig. 5b and Supplementary Figure 21). Figure 5c shows the volumetric capacitances of pseudocapacitor device as a function of scan rate. At 5 mV s$^{-1}$, the NP $c$-$V_2O_3$/$r$-$VO_{2-x}$ electrode reaches the highest volumetric capacitance of ~626 F cm$^{-3}$. As the scan rate is increased by 20 times (from 5 to 100 mV s$^{-1}$), the capacitance maintains ~77% (~487 F cm$^{-3}$), much higher than supercapacitors based on Au/$MnO_2$ multilayers[20] or graphene/$MnO_2$ hybrid electrodes[44]. Meanwhile, the Na$^+$ cation storage in the tunnels with $E_{int} = \sim -0.8$ eV essentially depresses the self-discharge of electrochemical cells. A voltage drop of ~0.006 V h$^{-1}$ in the symmetric NP $c$-$V_2O_3$/$r$-$VO_{2-x}$-

based pseudocapacitor is much lower than the ones with NP $r$-$VO_2$ (~0.016 V h$^{-1}$) and $c$-$V_2O_3$ (~0.026 V h$^{-1}$) electrodes, on which the charge storage is realized by surface redox reactions (Fig. 5d). The cycling life of the NP $c$-$V_2O_3$/$r$-$VO_{2-x}$ device is tested by galvanostatic charge/discharge at a current density of 130 A cm$^{-3}$ (inset of Fig. 5e). The significant capacitance retention, about 93% of the initial capacitance after 15,000 cycles (Fig. 5e), indicates its impressive long-term durability in the voltage window between 0 and 1.4 V.

The volumetric and gravimetric power and energy densities of the symmetric pseudocapacitor are calculated according to the volume and mass of NP $c$-$V_2O_3$/$r$-$VO_{2-x}$ in two electrodes, respectively. Their maxima reach ~330 mWh cm$^{-3}$ and ~320 mWh g$^{-1}$, very superior to the values reported in the double-layer supercapacitors based on onion-like carbon[5] or graphene/carbon nanotube hybrid[6], and the pseudocapacitors with electrode materials including pseudocapacitive MFLC-N[37], and conventional TMOs even with carbon nets or fibers, and nanoporous metal skeletons serving as conductive pathways (Fig. 5f and Supplementary Figure 22)[9, 19, 45]. When delivered at the maximum power of NP Au/$MnO_2$ pseudocapacitor and onion-like carbon supercapacitor (~280 W cm$^{-3}$)[5], our pseudocapacitor still has a volumetric energy density of ~110 mWh cm$^{-3}$, which is more than one and two orders of magnitude higher than their volumetric energies, respectively. The Ragone plot shown in Supplementary Figure 23 compares the volumetric power and energy based on the whole pseudocapacitor volume with those of commercially available energy-storage devices, where the volumes of two current collectors and one piece of paper separator are also

included in the calculation. The NP $c$-$V_2O_3$/$r$-$VO_{2-x}$-based pseudocapacitor stores charge with density of ~13 mWh cm$^{-3}$, which is slightly higher than that of 4 V/500 μAh thin-film lithium batteries[5, 6]. Furthermore, it can deliver high levels of electrical power, comparable to carbon-based supercapacitors[6]. These properties certify the unique capability of NP $c$-$V_2O_3$/$r$-$VO_{2-x}$-based pseudocapacitors to realize high-density energy storage/delivery at high power or fast charge/discharge rates, which makes them potentially competitive against batteries, such as lead-acid batteries and thin-film lithium batteries, for many high-power applications.

## Discussion

We have demonstrated 3D hierarchical NP $c$-$V_2O_3$/$r$-$VO_{2-x}$ films, which are fabricated by a facilely thermal-oxidation-actuated CTR-phase transformation of NP $c$-$V_2O_3$ precursor, as bipolar electrode materials for symmetric wide voltage window pseudocapacitors in aqueous electrolyte. The phase transformation enlists the in situ grown $r$-$VO_{2-x}$ layer to be composed of highly Na$^+$ accessible hexagonal oxygen-deficiency tunnels sandwiched between highly conductive $r$-$VO_2$ slabs, which essentially boosts the kinetics of redox and intercalation pseudocapacitance. Associated with electrode architecture of 3D bicontinuous and multimodal nanoporosity, the gravimetric and volumetric pseudocapacitances of the NP $c$-$V_2O_3$/$r$-$VO_{2-x}$ electrodes are enhanced relative to the nanoporous pristine $r$-$VO_2$ by a factor of 5–21 in a wide range of scan rates from 5 to 1000 mV s$^{-1}$. This renders their symmetric pseudocapacitor to reach a maximum volumetric energy of ~330 mWh cm$^{-3}$ (~13 mWh cm$^{-3}$ based on the whole volume of device, beyond that of 4 V/500 μAh thin-film lithium batteries) while delivering power densities similar to those of carbon-based supercapacitors. Furthermore, in a wide voltage window of 1.4 V, the pseudocapacitor exhibits exceptional low self-discharge behavior and outstanding long-term durability.

## Methods

**Fabrication of nanoporous isomeric vanadium oxide electrodes.** All 3D multimodal nanoporous film electrodes were fabricated by using polystyrene (PS) opal templates with size of 0.4 cm × 0.4 cm × 2 μm, which were assembled by NH$_4^+$-terminated PS nanospheres with diameter of ~450 nm on stainless steel substrates via evaporative deposition at 80 °C. An electrodeposition technique was employed to incorporate vanadium oxide into the PS opal templates on a three-electrode setup, in which a Pt foil and an Ag/AgCl electrode were used as the counter electrode and reference electrode. Following the electrodeposition at 1.5 V (vs Ag/AgCl) for 80 s in an electrolyte containing 1 M VOSO$_4$ and 1 mM H$_2$SO$_4$, NP $c$-$V_2O_3$ films with loading mass of ~20 and 130 μg were obtained by calcining the mixture films at 450 °C in H$_2$/Ar air, which enables selective removal of PS nanospheres and thermal reduction of vanadium oxide. Employing the as-prepared NP $c$-$V_2O_3$ films as the precursor scaffolds, NP $c$-$V_2O_3$/$r$-$VO_{2-x}$ films were prepared by thermal oxidation at 300 °C in a tube furnace sealed with ambient atmosphere, wherein the $r$-$VO_{2-x}$ layers were tuned by controlling the heat treatment time ranging from 2 to 60 min. The NP $r$-$VO_2$ films were achieved by extending the heat treatment time to 90 min.

**Structure characterizations.** The microstructures of NP $c$-$V_2O_3$, $c$-$V_2O_3$/$r$-$VO_{2-x}$ and $r$-$VO_2$ were investigated using a field-emission scanning microscope (JEOL JSM-6700F, 15 keV) and a field-emission transmission electron microscope (JEOL JEM-2100F, 200 keV). HR-STEM characterization was performed on a field-emission transition electron microscope (JEM-ARM200F, 200 kV) equipped with double spherical aberration correctors for the condenser lens and objective lens. The chemical composition was characterized by X-ray photoelectron spectroscopy on Thermo ECSALAB 250 with an Al anode. Binding energies were calibrated using containment carbon (C 1$s$ = 284.6 eV). Nitrogen adsorption/desorption isotherms at 77 K were measured on a micromeritics ASAP 2020 system to evaluate the specific surface area by the BET method, as well as the pore volume and the pore size by the Barrett–Joyner–Halenda (BJH) method. X-ray diffraction measurements were performed on a D/max2500pc diffractometer using Cu Kα radiation. Raman spectra were collected using a micro-Raman spectrometer (Renishaw) with a laser of 532 nm wavelength at 0.2 mW. Temperature dependence of resistivity was collected on Hall Effect measurement system (HMS-5000).

**Electrochemical measurements.** Electrochemical properties of single electrode were characterized in a classic three-electrode setup (Iviumstat electrochemical analyser, Ivium Technology) using Pt foil as counter electrode, Ag/AgCl electrode as reference electrode and 1 M Na$_2$SO$_4$ as aqueous electrolyte. Electrochemical performances of pseudocapacitor devices were measured in a two-electrode configuration. Cyclic voltammetry and galvanostatic charge/discharge were performed in potential windows from 0 to 0.8 V, −0.8 to 0.8 V and from 0 to 1.4 V at various scan rates and current densities, respectively. Self-discharge measurements were performed by charging pseudocapacitors based on NP $c$-$V_2O_3$, $c$-$V_2O_3$/$r$-$VO_{2-x}$ and $r$-$VO_2$ electrodes to 1.4 V at 0.2 mA, followed by open-circuit potential self-discharging for 50 h.

**DFT simulation.** All DFT computations were performed using the CASTEP code with ultrasoft pseudopotentials. The exchange-correlation effects were described by the generalized gradient approximation with the Perdew–Burke–Ernzerhof functional (PBE). A 400 eV cutoff was employed for the plane-wave basis set and the $k$-point separations in Brillouin zone were set as 0.04 Å$^{-1}$. The geometry optimizations were carried out until energy, maximum force and displacement were less than 10$^{-5}$ eV/atom, 0.03 eVÅ$^{-1}$ and 0.001 Å, respectively. The bulk phases of $VO_{2-x}$ ($x$ = 0.167 and 0.250) were simulated by optimizing both lattice parameters and atomic positions of $r$-$VO_2$, where an O atom was removed in the 3 × 1 × 1 and 2 × 1 × 1 supercells, respectively. Interface models consisting of $r$-$VO_2$(011) slabs (13.66 × 5.37 Å) and $c$-$V_2O_3$(012) slabs (14.85 × 5.47 Å) was established to simulate the coherent interface structures of $c$-$V_2O_3$/$r$-$VO_2$ and $c$-$V_2O_3$/$r$-$VO_{2-x}$ ($x$ = 0.167) with ordered or disordered oxygen vacancies after adding 20 Å-thick vacuum along the direction perpendicular to the interface. The intercalation energies ($E_{int}$) of a Na atom into the bulk $r$-$VO_2$, $r$-$VO_{2-x}$ ($x$ = 0.167 and 0.250) were calculated according to the equation: $E_{int} = E_{VO+Na}-(E_{VO} + 1/2E_{Na})$, where $E_{VO+Na}$, $E_{VO}$ and $E_{Na}$ are the total energies of vanadium oxides with one adsorbed Na atom, vanadium oxides and Na bulk, respectively. The energy barriers for Na diffusion in $VO_2$ and $VO_{2-x}$ were determined by optimizing several images between the initial and final structures along each path.

**Data availability.** All relevant data are available from the corresponding authors (xylang@jlu.edu.cn and jiangq@jlu.edu.cn) upon request.

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

## Acknowledgements

This work was supported by the National Natural Science Foundation of China (No. 51631004, 51422103), Top-notch Young Talent Program of China (W02070051), Chang Jiang Scholar Program of China (Q2016064), the Program for JLU Science and Technology Innovative Research Team (JLUSTIRT, 2017TD-09), the Fundamental Research Funds for the Central Universities and the Program for Innovative Research Team (in Science and Technology) in University of Jilin Province.

## Author contributions

X.-Y.L. and Q.J. conceived and designed the experiments. B.-T.L., X.-Y.L., L.G. and Z.W. carried out the fabrication of materials and performed the electrochemical and micro-structural characterizations. X.-M.S. and Q.J. performed the DFT calculations. X.-Y.L., Q. J. and B.-T.L. wrote the paper, and all authors discussed the results and commented on the manuscript.

## Additional information

**Competing interests:** The authors declare no competing interests.

