## [Peer Review File(PDF 896 kb) · Nature Communications]

Reviewers' comments:

Reviewer #1 (Remarks to the Author):

In this paper, the authors prepared corundum-to-rutile phase transformation vanadium oxides by controlled thermal-oxidation of porous V₂O₃ electrode. High diffusion-controlled capacity can be achieved with heterogenous c-V₂O₃/r-VO_{2-x} film. Although several nanometers metastable VO_{2-x} coatings were introduced, the surface states must change when exposed in aqueous solution, especially for vanadium oxides for which one of major problems is the change in oxidation states and vanadium dissolution during cycles. The significance of VO_{2-x} coating is very limited. The electrochemical rate tolerance is inferior to most of the reported capacitive electrodes, so the mentioned mechanism of pseudocapacitance is not reliable because of the diffusion-controlled reaction behavior in Fig. 3c. The weakness and incompleteness make it difficult for the reader to understand the significance of this work. Together with more discrepancies mentioned below, I do not recommend acceptance of this manuscript for publication in Nat. Commun.

More specific comments are provided below.

Question 1: The calculation methods of energy and power densities are missing. The mass loading and thickness of the active materials are also missing. These are essential information for the initial judgement of the performance. The calculation should contain the mass/volume from both active materials, substrates, and the packaging shell.

Question 2: The significance of the metastable VO_{2-x} coatings is questionable. Further V oxides coating cannot solve the known problem of chemical instability. Three electrode cycling test of the c-V₂O₃/r-VO_{2-x} should be presented in the manuscript. Importantly, the surface valence states after cycling should be analyzed.

Question 3: It is not accurate to call V₄O₇ phase (VO_{2-x}, x=1/4) in the whole manuscript. Firstly, it is not V₄O₇ phase in the material and V₄O₇ phase cannot be detected in the XRD Fig. 1c. It is most likely a heterogeneous structure with only VO₂ and V₂O₃ phases. Also, it is not accurate to evaluate the x value by XPS result of V³⁺/V⁴⁺ in this situation. Similarly, the whole crystalline structure in the composite structure of V₂O₃ and VO₂ cannot be determined by just one HRTEM image in Fig. 1f.

Question 4: Please recheck the calculation of capacity for the three-electrode measurement and the symmetric full cells. There should be a ~1/4 factor in their capacities, but the two systems in this paper have similar capacity values.

Question 5: The rate performance is poor in Fig. 3b. The capacities decrease dramatically with the increase of scanning rates. Fig. 3c shows the energy storage mechanism is the diffusion-controlled reaction (currents scale with $v^{1/2}$), not capacitive behavior. So the declaration of pseudocapacitive in the whole manuscript is totally contradictory with the obvious diffusion controlled battery behavior. For better understanding the difference, authors may want to refer to the Science paper, "Where Do Batteries End and Supercapacitors Begin?" by Patrice Simon et al.

Reviewer #2 (Remarks to the Author):

Despite high theoretical specific capacitance as the electrode for pseudo-capacitors, transition-metal oxides still face grand challenges to improve their performance because of their poor electron conductivity and limited redox reaction sites compared with carbon materials. In this work, Liu Bo-Tian et al fabricated 3D hierarchical NP c-V₂O₃/r-VO_{2-x} films through a thermal-oxidation process. This composite delivered an excellent specific capacitance around 1332 F g⁻¹ at a scan rate of 5mv/s in 1M Na₂SO₄ electrolyte. The performance of r-VO_{2-x} layer was well-studied and demonstrated using DFT simulation. However, there are still some aspects the author has to clarify:

- (1) In Page 5 line 99, the author claimed in the c-V₂O₃/r-VO_{2-x} composite, the former serves as the conductor for the electrons transport while the latter accommodates the cations storage. However, in Figure S1c, the resistance of the c-V₂O₃ film is clearly higher at room temperature (298K) than the c-V₂O₃/r-VO_{2-x} composite. How could the c-V₂O₃ interiors act as the conductor for the electrons transport? The electric properties showing in Figure S1c is also inconsistent with the I-V curves in Figure 2f.
- (2) Coordinates (a,b,c or x,y,z) need to be provided for Figure 1a,d,e,f. HAADF image in Figure 1f still showed atoms sitting inside the so-called "Hexagonal tunnels" of r-V₄O₇ phase. More information is needed to confirm the existence of the r-V₄O₇ phase.
- (3) Figure S1d,e were discussed after Figure S3a. Considering rearrange the images.
- (4) Important simulation/calculation results (Figure S1d,e; Figure S4 c,d; Figure S5, etc.) would be better to sit in the main manuscript (Figure 1) instead of the supporting information. Figure S6, Figure S8b-c, and Figure 1c-f can be combined and put in the main manuscript.
- (5) CV curves of NP c-V₂O₃/r-VO_{2-x} film electrodes with x = 0.22, 0.18, 0.14, and 0.10 were measured experimentally in a potential window of -0.8V ~ 0.8V. Can the author provide experimental results for x > 0.25 (expect c-V₂O₃) to compare with the theoretical predictions in Figure S5?

Response To Reviewers' Comments

Reviewer #1 (Remarks to the author):

In this paper, the authors prepared corundum-to-rutile phase transformation vanadium oxides by controlled thermal-oxidation of porous V_2O_3 electrode. High diffusion-controlled capacity can be achieved with heterogeneous c - V_2O_3/r - VO_{2-x} film.

Reply: We appreciate the reviewer for finding potential interest of our work and for the constructive comments. Following his/her suggestions and comments, we have supplemented some experiments, including three-electrode electrochemical characterization for all vanadium oxide specimens, XPS characterization of NP c - V_2O_3/r - VO_{2-x} ($x = 0.22$) electrode after 10,000 galvanostatic charge/discharge cycles; and UV-Visible spectroscopy measurements of the corresponding electrolyte; HAADF-STEM image simulation and analysis; I-V curves and resistivity-temperature curves]. Based on these new results that demonstrate not only the exceptional stability but also the unique pseudocapacitive mechanism, we have completely revised the manuscript. The detailed corrections are listed below.

Although several nanometers metastable VO_{2-x} coatings were introduced, the surface states must change when exposed in aqueous solution, especially for vanadium oxides for which one of major problems is the change in oxidation states and vanadium dissolution during cycles. The significance of VO_{2-x} coating is very limited. The electrochemical rate tolerance is inferior to most of the reported capacitive electrodes, so the mentioned mechanism of pseudocapacitance is not reliable because of the diffusion-controlled reaction behavior in Fig. 3c. The weakness and incompleteness make it difficult for the reader to understand the significance of this work. Together with more discrepancies mentioned below, I do not recommend acceptance of this manuscript for publication in Nat. Commun. More specific comments are provided below.

Reply: We agree with the reviewer that vanadium oxides usually suffer from electrochemical instability and thus fast capacitance fading during cycles because of vanadium dissolution in aqueous electrolyte. Furthermore, they also exhibit practical specific capacitance far below their theoretical values in energy storage. These problems seriously limit their wide applications in practical energy storage devices. In the present work, we demonstrate c - V_2O_3/r - VO_{2-x} heterogeneous electrodes with 3D bicontinuous and multimodal nanoporous architecture to exhibit superior pseudocapacitive behaviors in high-density charge storage, high rate performance and high stability, which result from the formation of r - VO_{2-x} layer via *in-situ* corundum-to-rutile phase transformation. To correctly illustrate the superior electrochemical

properties of the NP $c\text{-V}_2\text{O}_3/r\text{-VO}_{2-x}$ electrodes, we have carried out supplementary electrochemical measurements in a three-electrode configuration in the aqueous electrolyte of 1 M Na_2SO_4 as, where Pt foil and Ag/AgCl electrode are employed as counter electrode and reference electrode, respectively. In sharp contrast with the poor pseudocapacitive behaviors previously reported for vanadium oxides, the NP $c\text{-V}_2\text{O}_3/r\text{-VO}_{2-x}$ electrodes exhibit exceptional stability during galvanostatic charge/discharge cycles (*see* supplementary **Figure S17a**). The impressively enhanced durability in aqueous electrolyte is expected to result from the uniform coating of $r\text{-VO}_{2-x}$ layer with the coexistence of multivalent V, *i.e.*, V^{3+} and V^{4+} (**Figure 3c** and supplementary **Figure S9**). This is further confirmed by XPS characterization of the NP $c\text{-V}_2\text{O}_3/r\text{-VO}_{2-x}$ electrode after 10,000 cycles (supplementary **Figure S17b**), where there is only slight and negligible change of the x value from $x = 0.22$ to 0.23 even after repeated Na^+ intercalation/de-intercalation in aqueous electrolyte. Moreover, the UV-Visible spectroscopy measurements of the aqueous electrolyte before and after 10,000 cycles do not show remarkable change (supplementary **Figure S17c**), which implies that the long-term electrochemical cycles do not give rise to serious dissolution of vanadium and in turn verifies the exceptional stability of our NP $c\text{-V}_2\text{O}_3/r\text{-VO}_{2-x}$ electrodes. Meanwhile, the metastable $r\text{-VO}_{2-x}$ layer possesses not only the large-size quasi-hexagonal tunnels, which facilitates high Na^+ accessibility for more energy storage, but also the highly conductive V-V chains, which significantly improve the electron transport from the redox reaction sites to current collectors. These proper features enlist the NP $c\text{-V}_2\text{O}_3/r\text{-VO}_{2-x}$ electrodes to show much higher gravimetric and volumetric capacitances than the pristine NP $c\text{-V}_2\text{O}_3$ and $r\text{-VO}_2$ electrodes (**Figure 4a** and supplementary **Figure S12**). Even when the scan rate increasing to 1000 mV s^{-1} , the gravimetric and volumetric capacitances of the NP $c\text{-V}_2\text{O}_3/r\text{-VO}_{2-x}$ electrodes are about twenty times the values of NP V_2O_3 and $r\text{-VO}_2$ (**Figure 4b**). These facts evidently validate the significance of the metastable $r\text{-VO}_{2-x}$ layer in the remarkably enhanced energy storage of the 3D vanadium oxides.

Based on the re-performed three-electrode electrochemical measurements, the pseudocapacitive mechanisms and energy-storage performances of the NP $c\text{-V}_2\text{O}_3/r\text{-VO}_{2-x}$ electrodes are renewedly described and discussed according to their CV behaviors. The detailed results are shown in **Figure 4** and supplementary **Figure S12-16**. **Figure 4b** presents the gravimetric and volumetric capacitance of the NP $c\text{-V}_2\text{O}_3/r\text{-VO}_{2-x}$ ($x = 0.22$) electrode at various scan rate. In this plot, not only the values of the pristine NP $c\text{-V}_2\text{O}_3$ and $r\text{-VO}_2$ but also the highest values previously reported for typical pseudocapacitive electrode materials, such as RuO_2 , MnO_2 , hydrogenated- $\text{TiO}_2/\text{MnO}_2$ hybrid, $\text{Ti}_3\text{C}_2\text{T}_x$ MXene, N-doped mesoporous few-layer carbon and hexagonal WO_3 , are included for comparison. It should be noted that all electrode materials listed in this

plot are electrochemically characterized in a three-electrode configuration. As shown in **Figure 4b**, the NP $c\text{-V}_2\text{O}_3/r\text{-VO}_{2-x}$ electrode achieves a gravimetric capacitance of as high as $\sim 1856 \text{ F g}^{-1}$ (corresponding to the volumetric capacitance of $\sim 1933 \text{ F cm}^{-3}$) at a scan rate of 5 mV s^{-1} . When the scan rate increasing to 1000 mV s^{-1} , it still retains the capacitance of 760 F g^{-1} or 792 F cm^{-3} , much higher the aforementioned representative electrode materials even at much lower scan rates. This indicates the superior rate-performance of pseudocapacitive NP $c\text{-V}_2\text{O}_3/r\text{-VO}_{2-x}$ electrode due to the coexisting mechanisms of both intercalation and surface redox reaction, which are diffusion-controlled and surface-controlled processes, respectively. The kinetics is identified and analyzed by the relationships of current (i) versus scan rate (v) (supplementary **Figure S12d**) and capacitance (C) versus $v^{1/2}$ (**Figure 4c**).

In addition, we have also completely revised according to additional comments and suggestions. The detailed corrections are listed below. After revision with these constructive comments and suggestions, we feel the paper becomes much more solid and stronger, as well as easier for readers to understand the significance of this work. We would like to thank the reviewer for his/her profound help in this work and promoting the scientific significance of this manuscript. In view of the novelty, scientific implications and excellent energy-storage performances of isomeric NP $c\text{-V}_2\text{O}_3/r\text{-VO}_{2-x}$ hybrid electrodes, as pointed out by Reviewer #2, “*This composite delivered an excellent specific capacitance around 1332 F g⁻¹ at a scan rate of 5mv/s in 1M Na₂SO₄ electrolyte. The performance of r-VO_{2-x} layer was well-studied and demonstrated using DFT simulation*”, we wish the reviewer could share our confidence and belief that the work reported in this paper deserves to be published in a high-impact journal, like *Nature Communications*.

(1) The calculation methods of energy and power densities are missing. The mass loading and thickness of the active materials are also missing. These are essential information for the initial judgment of the performance. The calculation should contain the mass/volume from both active materials, substrates, and the packaging shell.

Reply: Following this suggestion, we have given a detailed description on the calculation method of energy and power densities in supplementary information. In addition, the loading mass ($20 \mu\text{g}$) and thickness ($\sim 1.2 \mu\text{m}$) of the vanadium oxides have been added in the text. To compare the gravimetric/volumetric capacitances of NP $c\text{-V}_2\text{O}_3/r\text{-VO}_{2-x}$ with the previously reported typical electrode materials, these values are calculated on the basis of the mass/volume of active material in single electrode (**Figure 4b**). While for the energy and power densities of pseudocapacitor devices (supplementary **Figure S21**), in which the calculation includes the volume from all parts of devices, such as active materials, substrates and packaging shell.

(2) The significance of the metastable VO_{2-x} coatings is questionable. Further V oxides coating cannot solve the known problem of chemical instability. Three electrode cycling test of the $c-V_2O_3/r-VO_{2-x}$ should be presented in the manuscript. Importantly, the surface valence states after cycling should be analyzed.

Reply: We thank the reviewer for this comment and suggestion. According to the suggestion, we have carried out supplementary electrochemical durability measurement of NP $c-V_2O_3/r-VO_{2-x}$ ($x = 0.22$) electrode in 1 M Na_2SO_4 aqueous electrolyte in a classic three-electrode configuration, where Pt foil and Ag/AgCl electrode are employed as counter electrode and reference electrode, respectively. During the 10,000 cycles of galvanostatic charge/discharge (inset of supplementary **Figure S17a**), the NP $c-V_2O_3/r-VO_{2-x}$ electrode exhibits exceptional capacitance retention (supplementary **Figure S17a**). XPS characterization of the NP $c-V_2O_3/r-VO_{2-x}$ electrode demonstrates that the repeated Na^+ intercalation/de-intercalation for 10,000 cycles only give rise to slight change of chemical state from $x = 0.22$ to 0.23. Moreover, UV-Visible spectra of the aqueous electrolyte, in which the long-term cycles of the NP $c-V_2O_3/r-VO_{2-x}$ electrode are performed, do not remarkably change during the cycles, implying that there is not too much vanadium to dissolve in the electrolyte. These indicate the high stability of the NP $c-V_2O_3/r-VO_{2-x}$ electrode in energy storage.

(3) It is not accurate to call V_4O_7 phase (VO_{2-x} , $x=1/4$) in the whole manuscript. Firstly, it is not V_4O_7 phase in the material and V_4O_7 phase cannot be detected in the XRD Fig. 1c. It is most likely a heterogeneous structure with only VO_2 and V_2O_3 phases. Also, it is not accurate to evaluate the x value by XPS result of V^{3+}/V^{4+} in this situation. Similarly, the whole crystalline structure in the composite structure of V_2O_3 and VO_2 cannot be determined by just one HRTEM image in Fig. 1f.

Reply: According to this comment, we have corrected the presentation of the new phase as the $r-VO_{2-x}$ in view that V_4O_7 phase cannot be detected in XRD patterns and Raman spectra. Therefore, we present our electrodes as NP $c-V_2O_3/r-VO_{2-x}$ in the whole manuscript to illustrate the heterogeneous structure of $c-V_2O_3$ and $r-VO_{2-x}$. Therein, the former is highly conductive and thus facilitate electron transport in a 3D bicontinuous skeleton, and the latter is composed of quasi-hexagonal tunnels and rutile slabs, which play roles in energy storage, *i.e.*, accommodating Na ions and transferring electrons, respectively. On the basis of these corrections, it is reasonable to evaluate the x value according to the XPS result of V^{3+}/V^{4+} ratio considering that the V^{3+} concentration in the $r-VO_{2-x}$ layer corresponds to the formation of quasi-hexagonal tunnels.

The microstructure of NP $c-V_2O_3/r-VO_{2-x}$ hybrid electrodes have been clearly illustrated by TEM and HRTEM images shown in **Figure 2d,e,f** and supplementary **Figure S6b,c**. Aberration-corrected high-angle annular dark-field (HAADF) scanning

TEM (HAADF-STEM) is also employed to characterize the crystalline structure of r -VO_{2-x} surface layer. Viewed along the [001] axis, the r -VO_{2-x} layer is composed of alternating quasi-hexagonal tunnels and rutile slabs with [1×1] tunnels. This unique structure is clearly shown in a typical HAADF-STEM image (**Figure 2g**). It should be noted that the atoms sitting inside the quasi-hexagonal tunnels are due to the V atoms in the c -V₂O₃ substrate. Such structure is further confirmed by image simulations, in which bright spots correspond to atomic columns, and the dark hexagons and squares are due to the quasi-hexagonal tunnels and [1×1] tunnels in the rutile slabs, respectively (**Figure 2h**). A typical line profile demonstrates a large and aperiodic fluctuation around ~1 nm to validate the quasi-hexagonal tunnel that is produced in the CTR transformation via shearing movement (**Figure 2i**).

(4) Please recheck the calculation of capacity for the three-electrode measurement and the symmetric full cells. There should be a ~1/4 factor in their capacities, but the two systems in this paper have similar capacity values.

Reply: Following this suggestion and the supplementary three-electrode measurements of NP c -V₂O₃, r -VO₂ and c -V₂O₃/ r -VO_{2-x} electrodes, we have re-calculated the gravimetric and volumetric capacitances of single electrode according to their CV curves (**Figure 4a,b** and supplementary **Figure S12a,b,c, S15, S16**). While for the symmetric pseudocapacitors assembled with two NP c -V₂O₃/ r -VO_{2-x} ($x = 0.22$) electrodes, the volumetric capacitances of devices are re-calculated according to the total volume of two electrodes (**Figure 5c**). There is about 1/4 factor in their capacitance.

(5) The rate performance is poor in Fig. 3b. The capacities decrease dramatically with the increase of scanning rates. Fig. 3c shows the energy storage mechanism is the diffusion-controlled reaction (currents scale with $v^{1/2}$), not capacitive behavior. So the declaration of pseudocapacitive in the whole manuscript is totally contradictory with the obvious diffusion controlled battery behavior. For better understanding the difference, authors may want to refer to the Science paper; “Where Do Batteries End and Supercapacitors Begin?” by Patrice Simon et al.

Reply: We thank the reviewer for the suggestive comment. To correctly demonstrate the rate performance of NP c -V₂O₃/ r -VO_{2-x}, c -V₂O₃ and r -VO₂ electrodes, their gravimetric and volumetric capacitances at various scan rates from 5 to 1000 mV s⁻¹ are re-calculated according to their CV curves in supplementary three-electrode measurements. **Figure 4b** shows typical semi-log plots of gravimetric and volumetric capacitance versus scan rate (v) for NP c -V₂O₃/ r -VO_{2-x} ($x = 0.22$), in comparison with these for NP c -V₂O₃ and r -VO₂ electrodes. As a result of the remarkably enhanced Na⁺ accessibility of the quasi-hexagonal tunnels and the highly conductive rutile slabs in the r -VO_{2-x} layer, the gravimetric capacitance reaches as high as ~1856 F g⁻¹ at 5 mV s⁻¹

¹. Even when the scan rate increasing to 1000 mV s⁻¹, this full-oxide electrode still retains ~760 F g⁻¹, about twenty times the values of the pristine NP *c*-V₂O₃ and *r*-VO₂. This impressive rate performance enlists the NP *c*-V₂O₃/*r*-VO_{2-x} (*x* = 0.22) electrode to outperform not only volumetrically but gravimetrically some of the best pseudocapacitive electrodes in a full rate range reported previously: such as mesoporous RuO₂ nanotubes, hydrogenated-TiO₂/MnO₂ (H-TiO₂/MnO₂) hybrid, bare MnO₂, Ti₃C₂T_x MXene clay, N-doped mesoporous few-layer carbon (MFLC-N) and nanostructured hexagonal WO₃ (**Figure 4b**).

To illustrate the unique discharge kinetics, electrochemical analysis is re-carried out based on three-electrode measurements. The cathodic current (*i*) at 0.4 V is assumed to obey a power-law relationship of scan rate (*v*), *i.e.*, $i = av^b$, with *a* and *b* being adjustable values. The *b*-value of 0.5 or 1 indicates that the current is a diffusion- or surface-controlled process, respectively. In a ln(*i*)-ln(*v*) plot (supplementary **Figure S12d**), the NP *c*-V₂O₃/*r*-VO_{2-x} (*x* = 0.22) electrode possesses a *b*-value of 1 in the scan rate ranging from 5 to 100 mV s⁻¹. This implies the surface-controlled kinetics in the discharge time >8 s as a result of enhanced Na⁺ accessibility in quasi-hexagonal tunnels, of which the intercalation energy is -0.8 eV and the energy barrier for Na⁺ transport is 0.02 eV (**Figure 1c,d**). While for *v* > 100 mV s⁻¹, the *b*-value decreases to 0.76, which is due to the kinetic constraint of Na⁺ diffusion in addition to ohmic contribution revealed by electrochemical impedance spectroscopy (EIS) analysis in a frequency range from 100 kHz to 10 mHz (supplementary **Figure S14**). The relationship between the capacitance and scan rate can also demonstrates the rate-limiting step of charge-storage mechanism. a plot of normalized capacitance versus $v^{-1/2}$ is shown in **Figure 4c**. Similar to the observation in supplementary **Figure S12d**, there exhibit two distinct regions in $v < 100$ mV s⁻¹ and $v > 100$ mV s⁻¹, further validating the fact that the total specific capacitance (*C_s*) of NP *c*-V₂O₃/*r*-VO_{2-x} arises from the diffusion/*r*-controlled intercalation pseudocapacitance (rate-dependent component, $\kappa_2 v^{-1/2}$) coupled with the surface-controlled redox pseudocapacitance (rate-independent component, κ_1), *i.e.*, $C_s = \kappa_1 + \kappa_2 v^{-1/2}$. At scan rates below 100 mV s⁻¹, the extrapolated intercept to $v^{-1/2} = 0$ yields the redox pseudocapacitance that just accounts for about two fifths of total capacitance. This indicates that the charge storage is mainly contributed by the Na⁺ insertion/extraction. Whereas the intercalation pseudocapacitance is conventionally limited by solid-state diffusion in conventional TMOs, it improves significantly the energy-storage density without compromise of rate performance in the NP *c*-V₂O₃/*r*-VO_{2-x} electrodes because of the unique microstructure of *r*-VO_{2-x}, which offers both redox and intercalation pseudocapacitance with similarly facile kinetics. The dominance of Na⁺ intercalation/de-intercalation is further demonstrated by the fact that the NP *c*-V₂O₃/*r*-VO_{2-x} electrodes maintain similar pseudocapacitive behaviors and rate

performance while their volumetric capacitances increase with the x value, but are independent of the production amount of rutile phase in the corundum-to-rutile phase transformation (**Figure 4d** and supplementary **Figure S15**).

Reviewer #2 (Remarks to the author):

Despite high theoretical specific capacitance as the electrode for pseudo-capacitors, transition-metal oxides still face grand challenges to improve their performance because of their poor electron conductivity and limited redox reaction sites compared with carbon materials. In this work, Liu Bo-Tian et al fabricated 3D hierarchical NP c - V_2O_3 / r - VO_{2-x} films through a thermal-oxidation process. This composite delivered an excellent specific capacitance around 1332 F g⁻¹ at a scan rate of 5mv/s in 1M Na₂SO₄ electrolyte. The performance of r - VO_{2-x} layer was well-studied and demonstrated using DFT simulation. However, there are still some aspects the author has to clarify:

Reply: We thank the reviewer for his/her insightful and constructive comments and for finding our work of significant interest. Following his/her suggestions and comments we have comprehensively revised the manuscript. The details can be found below.

(1) In Page 5 line 99, the author claimed in the c - V_2O_3 / r - VO_{2-x} composite, the former serves as the conductor for the electrons transport while the latter accommodates the cations storage. However, in Figure S1c, the resistance of the c - V_2O_3 film is clearly higher at room temperature (298K) than the c - V_2O_3 / r - VO_{2-x} composite. How could the c - V_2O_3 interiors act as the conductor for the electrons transport? The electric properties showing in Figure S1c is also inconsistent with the I - V curves in Figure 2f.

Reply: According to this comment, we have re-measured the electric properties of NP c - V_2O_3 / r - VO_{2-x} film and the pristine NP c - V_2O_3 and r - VO_2 films, including their temperature-resistivity curves (supplementary **Figure S1c**) and I - V curves (**Figure 3e** and inset). As we can see, both the temperature-resistivity and I - V curves consistently indicate that at room temperature the NP c - V_2O_3 has a lower resistance than NP c - V_2O_3 / r - VO_2 and thus serves as the conductor for the electrons transport.

(2) Coordinates (a,b,c or x,y,z) need to be provided for Figure 1a,d,e,f. HAADF image in Figure 1f still showed atoms sitting inside the so-called “Hexagonal tunnels” of r - V_4O_7 phase. More information is needed to confirm the existence of the r - V_4O_7 phase.

Reply: We thank the reviewer for the suggestions and comments. Following the suggestion, we have added the coordinates in the atomic schematics (**Figure 1a,b,e**) and HRTEM images (**Figure 2e,f**).

While for the HAADF-STEM image, the atoms sitting inside the quasi-hexagonal tunnels of $r\text{-VO}_{2-x}$ layer should be due to the atom columns that correspond to the V atoms in the $c\text{-V}_2\text{O}_3$ substrate.

Considering the suggestion proposed by Reviewer #1, we have corrected the presentation of new phase in the corundum-to-rutile phase transformation as $r\text{-VO}_{2-x}$, which is composed of alternating quasi-hexagonal tunnels and rutile slabs with $[1\times 1]$ tunnels. This unique microstructure is directly observed from atomic-resolution HAADF-STEM image (**Figure 2g**). Further image simulations confirm that bright spots correspond to atomic columns, among which the dark hexagons and squares are due to the quasi-hexagonal tunnels and $[1\times 1]$ tunnels in the rutile slabs, respectively (**Figure 2h**). A typical line profile demonstrates a large and aperiodic fluctuation around ~ 1 nm to validate the quasi-hexagonal tunnel that is produced in the corundum-to-rutile phase transformation via shearing movement (**Figure 2i**).

(3) *Figure S1d,e were discussed after Figure S3a. Considering rearrange the images.*

Reply: Following this suggestion, we have rearranged the images and moved the plots in supplementary **Figure S1d,e** into **Figure 1c,d**.

(4) *Important simulation/calculation results (Figure S1d,e; Figure S4 c,d; Figure S5, etc.) would be better to sit in the main manuscript (Figure 1) instead of the supporting information. Figure S6, Figure S8b-c, and Figure 1c-f can be combined and put in the main manuscript.*

Reply: According to this suggestion, we have rearranged **Figure 1,2,3** by moving supplementary **Figure S1d,e** and **Figure S4c,d** into **Figure 1c,d** in the main manuscript and combining supplementary **Figure S6** and **Figure S8b,c** with **Figure 1d-f** in **Figure 2** in the main manuscript. While for the supplementary **Figure S5**, we still put it in supplementary information in view that it presents the result similar to that in **Figure 4d**.

(5) *CV curves of NP $c\text{-V}_2\text{O}_3/r\text{-VO}_{2-x}$ film electrodes with $x = 0.22, 0.18, 0.14,$ and 0.10 were measured experimentally in a potential window of $-0.8V \sim 0.8V$. Can the author provide experimental results for $x > 0.25$ (expect $c\text{-V}_2\text{O}_3$) to compare with the theoretical predictions in Figure S5?*

Reply: Following this suggestion, we have performed three-electrode electrochemical measurements on the specimens with thermal-oxidation time of 2 and 6 minutes, *i.e.*, $x = 0.428$ and 0.337 (supplementary **Figure S16c,d**). Their volumetric capacitances are calculated according to the CV curves and shown in **Figure 4d** for comparing with the theoretical predictions.

Reviewers' comments:

Reviewer #1 (Remarks to the Author):

Authors responded to both referees' comments with lengthy arguments and new supporting data. Authors have conducted new measurements and incorporated quite substantial amount of new data and corrected some ambiguities, which indeed have strengthened their argument of a mixture of two charge storage mechanisms (a MnO₂-type diffusion intercalation and RuO₂-type surface redox) due to the mixture of two crystalline phases. The mixing phases are well characterized by XPS and STEM; the new data from three-electrode measurements give more evidence of the mixed storage mechanisms. On one hand, the whole idea of mixing two capacitive mechanisms in a mixed crystalline phase is not greatly new nor appealing – basically people know that very well. However, the radically increased capacitance of the mixture phases is terrific. Given the high popularity of vanadium oxide materials, the high quality figures, and completeness of the data in the revised manuscript, I would like to give them a chance. I don't know if there are comments from a third referee, I don't have further critical comments.

Most of the references are older than 3 years. But there are some more recent publications in 2017 on energy storage of vanadium oxide nanostructures (either pseudocapacitor or Na-ion battery) – two AEM Reviews. Authors need make a more thorough literature survey.

Reviewer #2 (Remarks to the Author):

The authors have addressed the comments raised by the reviewers with adequate support and detailed discussion. I have no further comments and I believe it is publishable in its current form.

Reviewer #3 (Remarks to the Author):

In this work, the 3D hierarchical NP c-V₂O₃/r-VO₂-x films are fabricated by a facilely thermal-oxidation-actuated corundum-to-rutile phase transformation of NP c-V₂O₃ precursor. This strategy essentially boosts the kinetics of redox and intercalation pseudocapacitance. As bipolar electrode materials for symmetric pseudocapacitors in aqueous electrolyte, the c-V₂O₃/r-VO₂-x films display excellent volumetric-capacitance performances. This work is interesting, and opens up a promising route to improve the capacitance performance of TMO materials. However, there are many questions that the authors did not give reasonable explanation; some electrochemical tests and calculation are not convincing. The authors need to revise and resubmit the manuscript before it is suitable for publication in this journal.

1. The r-VO₂-x has a tunnel structure, in which the cations prefer to transport along the z-axis tunnel. What's the size and morphology of the tunnel?
2. In the c-V₂O₃/r-VO₂-x films, what's the proportion of r-VO₂-x?
3. In the process of experiment, NP c-V₂O₃ films were obtained by calcining the mixture films at 450 °C in H₂/Ar air. However, the PS will be converted to be carbon. What is ratio of carbon? The carbon also helps to improve the electrical conductivity and capacitance. How can differentiate the double layer capacitance of carbon with that of VO_x?
4. Figure 4a shows a quasi-rectangular shape in a potential window of 0 to 0.8 V that measured in a three-electrode configuration. Based on the symmetric pseudocapacitors, their voltage windows can be extended to 1.4 V from 0.8 V. Why? During the process of charge/discharge, how the potential of positive and negative electrodes change?

5. In the experimental section, NP c-V₂O₃ films with loading mass of ~20 μg were obtained. How to determine the weight accurately? In addition, the mass loading is too small, much large mass loading (>500 μg/cm²) may be more persuasive, thus is necessary to add.
6. In Figure 5, the authors demonstrated that the cation intercalation kinetics play an important role in improving the capacitance performance. Ex situ XRD spectra of electrodes intercalated with cations is a direct method to prove the intercalation-mechanism.
7. In this work, the maxima volumetric energy density can be up to ~330 mWh cm⁻³. But I doubt that. Please confirm that your calculation is correct.

Response To Reviewers' Comments

Reviewer #1 (Remarks to the Author):

Authors responded to both referees' comments with lengthy arguments and new supporting data. Authors have conducted new measurements and incorporated quite substantial amount of new data and corrected some ambiguities, which indeed have strengthened their argument of a mixture of two charge storage mechanisms (a MnO₂-type diffusion intercalation and RuO₂-type surface redox) due to the mixture of two crystalline phases. The mixing phases are well characterized by XPS and STEM; the new data from three-electrode measurements give more evidence of the mixed storage mechanisms. On one hand, the whole idea of mixing two capacitive mechanisms in a mixed crystalline phase is not greatly new nor appealing – basically people know that very well. However, the radically increased capacitance of the mixture phases is terrific. Given the high popularity of vanadium oxide materials, the high quality figures, and completeness of the data in the revised manuscript, I would like to give them a chance. I don't know if there are comments from a third referee, I don't have further critical comments.

Reply: We appreciate the Reviewer for his/her positive comments and recommendation for publication in *Nature Communications*.

Most of the references are older than 3 years. But there are some more recent publications in 2017 on energy storage of vanadium oxide nanostructures (either pseudocapacitor or Na-ion battery) – two AEM Reviews. Authors need make a more thorough literature survey.

Reply: Following the Reviewer's suggestion, we have made a more thorough literature survey and added necessary publications in the references.

Reviewer #2 (Remarks to the Author):

The authors have addressed the comments raised by the reviewers with adequate support and detailed discussion. I have no further comments and I believe it is publishable in its current form.

Reply: We appreciate the Reviewer for satisfying our revision and recommending it for publication in *Nature Communications*.

Reviewer #3 (Remarks to the Author):

In this work, the 3D hierarchical NP $c\text{-V}_2\text{O}_3/r\text{-VO}_{2-x}$ films are fabricated by a facilely thermal-oxidation-actuated corundum-to-rutile phase transformation of NP $c\text{-V}_2\text{O}_3$ precursor. This strategy essentially boosts the kinetics of redox and intercalation pseudocapacitance. As bipolar electrode materials for symmetric pseudocapacitors in aqueous electrolyte, the $c\text{-V}_2\text{O}_3/r\text{-VO}_{2-x}$ films display excellent volumetric-capacitance performances. This work is interesting, and opens up a promising route to improve the capacitance performance of TMO materials. However, there are many questions that the authors did not give reasonable explanation; some electrochemical tests and calculation are not convincing. The authors need to revise and resubmit the manuscript before it is suitable for publication in this journal.

Reply: We appreciate the Reviewer for his/her valuable comments. We also would like to thank him/her for the constructive comments and suggestions. Following these comments and suggestions, we have revised the manuscript. The detailed corrections are listed below.

(1) The $r\text{-VO}_{2-x}$ has a tunnel structure, in which the cations prefer to transport along the z -axis tunnel. What's the size and morphology of the tunnel?

Reply: As a result of the corundum-to-rutile (CTR) phase transformation in oxygen nonstoichiometry, the $r\text{-VO}_{2-x}$ layer is composed of quasi-hexagonal tunnels sandwiched between general rutile slabs. This unique tunnel structure is illustrated in schematic image (**Figure 1b**) and confirmed by typical STEM image (**Figure 2g**) and its simulated image (**Figure 2h**). Therein, the cross-section area of the quasi-hexagonal tunnels is $\sim 24 \text{ \AA}^2$, much larger than the value ($\sim 10 \text{ \AA}^2$) of [1 \times 1] tunnels in the general rutile slabs.

(2) In the $c\text{-V}_2\text{O}_3/r\text{-VO}_{2-x}$ films, what's the proportion of $r\text{-VO}_{2-x}$?

Reply: Considering that the volumetric capacitances of NP $c\text{-V}_2\text{O}_3/r\text{-VO}_{2-x}$ films increase with the x value, but are independent of the production amount of rutile phase, the proportion of $r\text{-VO}_{2-x}$ only in the NP $c\text{-V}_2\text{O}_3/r\text{-VO}_{2-x}$ film with a thermal oxidation time of 10 min has been determined according to the Rietveld refinement of XRD patterns. The refinement gives rise to the weight ratio of 43.2: 56.8 for $c\text{-V}_2\text{O}_3$: $r\text{-VO}_{2-x}$.

(3) In the process of experiment, NP $c\text{-V}_2\text{O}_3$ films were obtained by calcining the mixture films at 450 °C in H_2/Ar air. However, the PS will be converted to be carbon. What is ratio of carbon? The carbon also helps to improve the electrical conductivity and capacitance. How can differentiate the double layer capacitance of carbon with

that of VO_x ?

Reply: Following this comment, we have performed supplementary Raman characterization on the specimen of NP $c\text{-}V_2O_3/r\text{-}VO_{2-x}$ film with thermal oxidation of 10 min in an extended wavenumber region from 1000 to 1600 cm^{-1} , in which the characteristic Raman peaks of carbon generally appear. The Raman spectrum is shown in supplementary **Figure S8b**, where there are not any Raman peaks of carbon to be observed. It demonstrates that the selective removal of PS nanospheres by calcination does not give rise to the formation of carbon. This result is further confirmed by XRD patterns of all specimens, including NP $c\text{-}V_2O_3$, $r\text{-}VO_2$, and $c\text{-}V_2O_3/r\text{-}VO_{2-x}$ films (**Figure 3a**), where there is not any characteristic diffraction peak of carbon at $2\theta = \sim 26^\circ$. Therefore, the electrical conductivity and specific capacitance of the film electrodes are the intrinsic properties of vanadium oxide films, which are not influenced by carbon materials.

(4) *Figure 4a shows a quasi-rectangular shape in a potential window of 0 to 0.8 V that measured in a three-electrode configuration. Based on the symmetric pseudocapacitors, their voltage windows can be extended to 1.4 V from 0.8 V. Why? During the process of charge/discharge, how the potential of positive and negative electrodes change?*

Reply: Electrochemical measurements in the potential windows from 0 to 0.8 V (**Figure 4a** and supplementary **Figure S12, S13, S16**) and from -0.8 to 0.8 V (supplementary **Figure S19**) demonstrate that the vanadium oxides are bipolar electrode materials and can work as both negative and positive electrodes for pseudocapacitive energy storage, like carbon for capacitive energy storage in supercapacitors. When two identical NP $c\text{-}V_2O_3/r\text{-}VO_{2-x}$ electrodes are constructed into a full cell, one works as positive electrode and the other serves as negative electrode. This leads to a high working voltage of 1.6 V in the full device. For avoiding any overestimation of electrochemical properties (such as specific capacitance as well as energy and power densities) due to the oxygen evolution reaction, in this work, we only extend the voltage window to 1.4 V. During the process of charge/discharge, the potential change of positive and negative electrodes changes is similar to that taking place in the general electrodes in symmetric pseudocapacitors.

(5) *In the experimental section, NP $c\text{-}V_2O_3$ films with loading mass of $\sim 20 \mu\text{g}$ were obtained. How to determine the weight accurately? In additional, the mass loading is too small, much large mass loading ($> 500 \mu\text{g cm}^{-2}$) may be more persuasive, thus is necessary to add.*

Reply: We appreciate the Reviewer for his/her comment and suggestion. The mass of

$c\text{-V}_2\text{O}_3$ is determined by the weight change before and after loading $c\text{-V}_2\text{O}_3$. In addition, we have carried out supplementary experiment to prepare NP $c\text{-V}_2\text{O}_3/r\text{-VO}_{2-x}$ film with a higher loading mass (130 μg , or 810 $\mu\text{g cm}^{-2}$), which is obtained by increasing the thickness of film electrode (**Figure S14a**). Whereas the thickness of the NP $c\text{-V}_2\text{O}_3/r\text{-VO}_{2-x}$ film reaches 7.8 μm , it still exhibits the almost same pseudocapacitive energy-storage behaviors as the thinner one, as shown in supplementary **Figure S14b** and inset.

(6) In Figure 5, the authors demonstrated that the cation intercalation kinetics play an important role in improving the capacitance performance. Ex situ XRD spectra of electrodes intercalated with cations is a direct method to prove the intercalation-mechanism.

Reply: Following the Reviewer's constructive suggestion, we have performed ex situ XRD characterizations on the NP $c\text{-V}_2\text{O}_3/r\text{-VO}_{2-x}$ electrode before and after Na^+ intercalation and found that there are no apparent phase changes. This may be one of the reasons why our specimens exhibit exceptionally high stability in a long-term electrochemical cycling. Alternatively, we have carried out supplementary Raman spectrum measurements on the NP $c\text{-V}_2\text{O}_3/r\text{-VO}_{2-x}$ electrode before and after Na^+ intercalation. The detailed results are shown in supplementary **Figure S20**. Compared with the Raman spectrum of NP $c\text{-V}_2\text{O}_3/r\text{-VO}_{2-x}$ film without Na^+ intercalation, there is a new characteristic Raman peak at 962 cm^{-1} for the one with Na^+ intercalation. The presence of such line in the frequency region related to vanadyl stretching modes suggests strong interaction of sodium with the apical oxygen atoms in the $r\text{-VO}_{2-x}$ tunnels. In addition, the Na^+ intercalation also leads to new peak at 224 cm^{-1} and appearance of shoulders at 420 and 162 cm^{-1} .

(7) In this work, the maxima volumetric energy density can be up to $\sim 330 \text{ mWh cm}^{-3}$. But I doubt that. Please confirm that your calculation is correct.

Reply: Following this suggestion, we have double-checked our data and calculations. The maximum volumetric energy density of NP $c\text{-V}_2\text{O}_3/r\text{-VO}_{2-x}$ electrode can reach $\sim 330 \text{ mWh cm}^{-3}$.

REVIEWERS' COMMENTS:

Reviewer #3 (Remarks to the Author):

The authors have addressed the comments raised by the reviewers with detailed discussion. I have no further comments and it is publishable in its current form.

Response To Reviewer's Comments

Reviewer #3 (Remarks to the Author):

The authors have addressed the comments raised by the reviewers with detailed discussion. I have no further comments and it is publishable in its current form.

Reply: We appreciate the Reviewer for his/her positive comments and recommendation for publication in *Nature Communications*.